# The IAA-Producing Rhizobacterium *Bacillus* sp. SYM-4 Promotes Maize Growth and Yield

**DOI:** 10.3390/plants14111587

**Published:** 2025-05-23

**Authors:** Yumeng Song, Qifei Chen, Juan Hua, Shaobin Zhang, Shihong Luo

**Affiliations:** Engineering Research Center of Protection and Utilization of Plant Resources, College of Bioscience and Biotechnology, Shenyang Agricultural University, Shenyang 110866, China; 15524112155@163.com (Y.S.); huajuan@syau.edu.cn (J.H.)

**Keywords:** maize, PGPR, IAA, microbial fertilizer, *Bacillus* spp.

## Abstract

The application of microbial fertilizers derived from plant growth-promoting rhizobacteria (PGPR) is an important approach to increase crop yield while reducing the use of chemical fertilizers. Here, UPLC-MS/MS analyses were used to identify a bacterium, *Bacillus* sp. SYM-4, with a strong ability to secrete IAA. The strain was identified from 36 bacteria obtained from the rhizospheric soil of maize. Further inoculation experiments showed that *Bacillus* sp. SYM-4 was able to colonize the maize rhizosphere, resulting in a significant increase in IAA concentrations in seedlings. In addition, the antioxidant enzyme activity and chlorophyll content of maize seedlings were also significantly increased after inoculation with *Bacillus* sp. SYM-4. Therefore, *Bacillus* sp. SYM-4 was determined to be a PGPR for maize seedling growth. After further making it into microbial fertilizer, we found that, when 20% of the normal amount of chemical fertilizer was replaced with microbial fertilizer (*Bacillus* sp. SYM-4) and applied to field-cultivated maize seedlings, the growth of the maize plants at different stages was significantly promoted. Compared with the maize grown following application of a commercial microbial fertilizer (Pathfinder pioneer), seedlings grown using 20% *Bacillus* sp. SYM-4 microbial fertilizer and 80% chemical fertilizer showed significantly increased height. Substitution of chemical fertilizer (20%) with microbial fertilizer (*Bacillus* sp. SYM-4) treatment resulted in increases in maize yield over several measures (numbers of kernel rows on each ear, bald tip length, 100-grain weight and yield, and kernel nutrient content) compared to plants treated with pure chemical fertilizer. In this study, the replacement of a proportion of conventional fertilizer with a microbial substitute demonstrates a new technique with great potential for the green and efficient cultivation of maize.

## 1. Introduction

Fertilization is very important for improving yield in modern agricultural production [1]. Chemical fertilization is the most common agricultural measure to increase crop yield by improving soil nutrients [2]. However, problems such as high costs, destruction of soil structures, and groundwater pollution have resulted from the overuse of fertilizers [3,4]. At present, studies have shown that the use of biological fertilizers—products prepared by living microorganisms that can promote the growth of agricultural plants—can reduce the abuse of harmful chemical fertilizers [5,6]. At present, studies have shown that using biological fertilizers instead of some chemical fertilizers can improve soil physical and chemical properties, enrich organic matter, balance nutrients, regulate the structure and function of microbial communities, promote plant growth, increase crop yield, and enhance the resistance of plants to pathogens and pests [7]. Although several rhizosphere bacteria are considered to be biological fertilizers and biopesticides, they may have the ability to colonize human tissues and organs, leading to infection and disease development [8]. So, what kind of microorganisms are suitable as microbial fertilizers and how to apply microbial fertilizers scientifically are still unresolved questions.

As a staple food and feed crop, maize is the third largest crop in the world and is widely cultivated in China. The black soil in northeastern China is known for its high organic matter content and fertility, which is also indispensable for maintaining high maize yields [9]. However, in recent years, excessive fertilization has caused serious soil hardening, acidification, and nutrient imbalance in the soil, which has a negative impact on food security [10]. The fertility of black soil in Northeast China has decreased, and fertilization cannot maintain grain production [11]. At present, studies have shown that microbial fertilizer promotes crop yield and restores soil fertility by activating various microorganisms in the soil [12]. In addition, microbial fertilizer is also involved in the interaction with plant roots, improving the absorption of plant nutrients, thereby increasing crop yield [13]. For example, *Azotobacter chroococcum*, *Azospirillum brasilense*, and *Pseudomonas* spp. can be made into microbial fertilizer to promote maize growth [14,15]. Therefore, the replacement of chemical fertilizer with microbial fertilizer is the best choice at present.

At present, many bacterial fertilizers on the market are composed of one or more PGPR [16]. The bacteria most commonly used in the preparation of biological fertilizers are *Azospirillum* spp., *Bacillus* spp., and *Pseudomonas* spp. [17]. PGPR can increase the concentration of phytohormone IAA in plants by secreting phytohormone IAA, thereby achieving the effect of promoting plant growth [18]. For example, *Leclercia decarboxylata* MCH-1 and *Lysinibacillus* spp. can secrete large concentrations of IAA to promote the growth of maize [19]. In addition, IAA also plays an important role in bacterial chemotaxis [20]. At present, studies have shown that the use of PGPR to make microbial fertilizer combined with chemical fertilizer can help reduce the use of inorganic fertilizer and improve soil fertility. However, the optimal proportions of microbial fertilizer and chemical fertilizer application remain to be explored. This study isolated PGPR from maize rhizosphere soil, elucidated their growth-promoting mechanisms, developed microbial fertilizers, and determined the optimal chemical-to-microbial fertilizer ratio through systematic evaluation. Finally, a precision fertilization strategy was proposed to achieve sustainable intensification in agriculture.

## 2. Results

### 2.1. Qualitative and Quantitative Analyses of Bacterial IAA-Producing Ability and Determination of Movement Ability

The 36 strains of bacteria isolated and purified from the maize rhizosphere soil in this study were cultivated in medium-frequency transformer loam. Each strain was then inoculated into LB liquid medium for activation for 12 h to 48 h. The ability of each strain to synthesize IAA was qualitatively and quantitatively analyzed using methanol extraction followed by UPLC-MS/MS in MRM analysis mode. The results showed that three of the 36 bacterial strains had the ability to synthesize IAA, and these strains were named SYM-4, SYM-5, and SYM-6 (Appendix A). The UPLC-MS/MS analysis showed that SYM-4 had the strongest ability to synthesize IAA, and following 48 h growth, the concentration of IAA produced by this strain had reached the highest value observed in any strain (6.91 ± 0.57 μg·mL^−1^, Figure 1A). The second highest IAA concentration was produced by the SYM-6 strain (3.89 ± 0.06 μg·mL^−1^, Figure 1C). The bacterium with the weakest ability to synthesize IAA was the SYM-5 strain, the IAA produced by this strain reaching a concentration of 2.70 ± 0.62 μg·mL^−1^ (Figure 1B).

The ability of bacteria to move can help them rapidly colonize plant roots. The movement ability of the three IAA-producing strains of bacteria isolated in our experiment was determined (Figure 1D–F). The results demonstrated that strain SYM-4 showed significantly higher movement than either of the other two bacterial strains (4.98 ± 0.15 cm), while the maximum distance traveled by the other two strains lay between 3.3 and 3.7 cm (Figure 1G).

The ability of each of the different strains to solubilize potassium was also determined. Our results suggested that the ability of strain SYM-4 to solubilize potassium was also significantly higher than that of the other two bacterial strains (Appendix A). SYM-4 was therefore selected for follow-up study.

### 2.2. Identification the PGPR of Maize

The DNA of the SYM-4 bacterial strain was extracted and amplified using PCR. We next performed a BLAST sequence similarity search in the NCBI database to identify similar sequences. We then used our own data together with sequences downloaded from NCBI to construct a phylogenetic tree using the neighbor-joining method in MEGA 7. Support for the phylogenetic tree was based on 1000 bootstrap replicates of guided analysis. SYM-4 was supported as *Bacillus* coreaensis with 93% confidence, and SYM-4 was therefore identified as *Bacillus* sp. SYM-4 (Figure 2).

### 2.3. The Effect of Bacillus sp. SYM-4 on the Growth of Maize Seedlings

Maize seedlings were inoculated with *Bacillus* sp. SYM-4 and grown in pots, with seedlings treated with LB medium as the control group. After inoculation with *Bacillus* sp. SYM-4, the leaf area and stem diameter of maize seedlings were significantly higher than those of maize seedlings in the control group, with values of 57.78 ± 2.36 cm^2^ and 1.60 ± 0.17 cm in the inoculated group, respectively. Compared with the control group, maize seedlings inoculated with *Bacillus* sp. SYM-4 showed 56.01 ± 5.92% and 34.44 ± 9.08% increases in leaf area and stem diameter, respectively (Figure 3C,D).

The weights of shoots and roots of maize seedlings inoculated with *Bacillus* sp. SYM-4 were also significantly higher than those in the control group, showing an increase in the dry weights of shoots and roots of 41.62 ± 0.67% and 71.52 ± 0.79%, respectively, over those of the control (Figure 3E).

Plant height at different growth stages was also measured. We found that the difference in plant height between the treatment and control groups became more significant over time, with the difference between the two groups reaching the most significant level (*p* ≤ 0.001) at 16 d and 20 d following germination. The plant height reached 38.98 ± 1.29 cm and 46.2 ± 0.76 cm at 16 d and 20 d, respectively, which was significantly higher than that of the control groups (28.82 ± 3.36% and 32.09 ± 0.52%, Figure 3F).

### 2.4. Antioxidant Enzyme Activity, Chlorophyll Content, and IAA Concentration in Maize Seedlings Following Inoculation with Bacillus sp. SYM-4

The effect of inoculation with *Bacillus* sp. SYM-4 compared with the control was reflected in significant increases in POD, CAT, and SOD activities (42.84 ± 3.55%, 22.37 ± 1.59%, and 39.72 ± 4.03%; *p* ≤ 0.001) and differences in soluble protein and chlorophyll content, as well as MDA accumulation in pot-grown maize seedlings, after 14 dpi of treatment (Figure 4A–C). Compared with un-inoculated maize, the chlorophyll content of maize inoculated with *Bacillus* sp. SYM-4 also increased to 50.77 ± 2.61% (Figure 4D). The concentration of MDA in maize seedlings, however, decreased significantly (*p* ≤ 0.05) to 0.018 ± 0.001 μmol·L^−1^ and was 10.36 ± 2.38% lower than that of the control group (Figure 4E).

The concentration of the phytohormone IAA in maize seedlings was qualitatively and quantitatively analyzed using UPLC-MS/MS (Appendix A). We found that the IAA concentration in the shoots and roots of maize seedlings increased significantly at 7 d and 14 d following inoculation with *Bacillus* sp. SYM-4 compared with those of the control. After 7 d of treatment, the IAA concentrations of shoots and roots of maize were 1.39 ± 0.17 μg·g^−1^ FW and 0.62 ± 0.17 μg·g^−1^ FW, respectively (Figure 4G), increases of 77.19 ± 0.31% and 65.02 ± 8.28%, respectively, compared with the control group. At 14 d of treatment, the IAA concentrations of the shoots and roots were 0.63 ± 0.17 μg·g^−1^ FW and 0.15 ± 0.02 μg·g^−1^ FW, respectively, increases of 58.86 ± 6.52% and 61.20 ± 5.28%, respectively, compared with the control (Figure 4H).

### 2.5. The Effects of Replacement of Different Proportions of Chemical Fertilizer with the Bacillus sp. SYM-4 Microbial Fertilizer on the Growth of Maize Seedlings in the Field

In order to determine the best ratio of chemical to microbial fertilizer for the promotion of maize seedling growth, different proportions of chemical fertilizer and *Bacillus* sp. SYM-4 microbial fertilizer were applied to maize cultivated in the experimental field. The four key stages in the growth of maize seedlings, the seedling stage, the jointing stage, the big trumpet stage, and the tasseling stage, were assessed. Overall, replacement of 20% of chemical fertilizer with *Bacillus* sp. SYM-4 had the most significant effect of all treatments on the growth of maize plant height compared with the control (*p* ≤ 0.01). At the seedling, big trumpet, and tasseling stages, the application of fertilizer in which 20% chemical fertilizer had been replaced by *Bacillus* sp. SYM-4 microbial fertilizer resulted in significant promotion of plant height (*p* ≤ 0.001), and in the seedling and big trumpet stages, height increased by 34.09 ± 0.38% and 35.61 ± 6.09%, respectively, compared with the control (Figure 5A,C).

At the same time, the leaf area of maize at the four growth stages was measured under different fertilization treatments. Interestingly, the leaf area of maize seedlings was significantly higher than that of the other treatment groups (*p* ≤ 0.001) when the plants were treated with 40% *Bacillus* sp. SYM-4 microbial fertilizer and 60% chemical fertilizer, with the leaf area reaching 55.67 ± 1.07 cm^2^. The application of fertilizer with 40% *Bacillus* sp. SYM-4 microbial fertilizer led to a significant increase (52.15 ± 0.9%) in leaf area at the seedling stage (Figure 5E). However, the leaf area of maize seedlings treated with *Bacillus* sp. SYM-4 microbial fertilizer at 20% reached 992.46 ± 41.09 cm^2^, a significant increase compared with the other treatment groups at this stage (*p* ≤ 0.001) and an increase of 28.21 ± 3.41% compared to the control (Figure 5F). When the growth of maize entered the big trumpet stage, the leaf area of the group treated with fertilizer in which 20% chemical fertilizer was replaced with *Bacillus* sp. SYM-4 microbial fertilizer was the highest, reaching 3737.14 ± 134.71 cm^2^ (control: 2933.89 ± 177.97 cm^2^), an increase of 21.36 ± 7.04% compared with control group (Figure 5G). When the maize growth entered the tasseling stage, the leaf area values of each treatment group were all considerable, but the differences between them were small (between 6756.53 ± 87.63 cm^2^ and 7032.36 ± 132.72 cm^2^) and not significant (Figure 5H).

### 2.6. Treatment with Bacillus sp. SYM-4 Microbial Fertilizer Significantly Enhances Maize Height and Leaf Area Compared to Commercial Microbial Fertilizers

The height of maize at each of the four growth stages was measured following application of fertilizer in which different proportions of a chemical fertilizer were replaced with either *Bacillus* sp. SYM-4 or a commercial microbial fertilizer. In the treatment group in which chemical fertilizer had been replaced with *Bacillus* sp. SYM-4 microbial fertilizer, the promotional effect on the maize plant height at all of the four stages was significantly higher than in those plants treated with a proportion of commercial microbial fertilizer. At the jointing stage and the big trumpet stage, plants that received a fertilizer containing 20% *Bacillus* sp. SYM-4 showed plant heights reaching 84.1 ± 1.59 cm and 177.04 ± 7.93 cm, respectively, which were significantly higher (62.91 ± 0.71% and 52.42 ± 2.61%) than the maize treated with the same proportion of commercial microbial fertilizer.

The leaf area of the maize plants at each of the four growth stages was also measured. At the seedling and jointing stages, the promotion of leaf area after the chemical fertilizer was replaced with *Bacillus* sp. SYM-4 microbial fertilizer was significantly higher than the promotion achieved by the commercially available product at the same proportion (Appendix A).

### 2.7. The Effects of Replacement of Different Proportions of Chemical Fertilizer with Bacillus sp. SYM-4 Microbial Fertilizer on Maize Yield

The physiological indexes of maize were measured following treatment of the plants with different proportions of chemical and microbial fertilizer (Table 1). We found that the growth traits and yield of maize in all treatment groups were significantly different from those in the control. The panicle (cob) length in maize plants treated with 60% *Bacillus* sp. SYM-4 and 40% chemical fertilizer increased significantly (31.42 ± 2.19%, *p* ≤ 0.001, reaching 21.11 ± 2.23 cm) compared with the control group. Panicle diameter varied between 4.75 ± 0.23 cm and 4.99 ± 0.19 cm among the different treatments. The bald tip of the panicle could be significantly reduced by applying *Bacillus* sp. SYM-4 microbial fertilizer in any proportion, although the treatment with 20% *Bacillus* sp. SYM-4 microbial fertilizer was the most significant (*p* ≤ 0.001), where the length of the bald tip length decreased by 72.38 ± 3.59% compared with that of the control. The number of rows of kernels per ear increased most dramatically in plants treated with 40% *Bacillus* sp. SYM-4 microbial fertilizer and 60% chemical fertilizer, reaching 16.36 ± 1.75 lines, an increase of 14.05 ± 4.49% compared with the control group.

Plants treated with fertilizer comprising 20% *Bacillus* sp. SYM-4 microbial fertilizer and 80% chemical fertilizer showed the most significant increases in number of kernels per row and 100-grain weight (*p* ≤ 0.001), reaching values of 39.92 ± 3.48 kernels and 34.49 ± 0.38 g, respectively.

The yield of maize in kg was then calculated in plants treated with different proportions of *Bacillus* sp. SYM-4 microbial fertilizer and chemical fertilizer. Where the fertilizer contained 20% *Bacillus* sp. SYM-4 microbial fertilizer, the yield showed a significant increase, reaching 883.18 ± 13.17 kg/mu, an increase of 32.89 ± 0.94% compared to the control. Based on the above results, we concluded that the application of chemical fertilizer in which 20% had been replaced with *Bacillus* sp. SYM-4 microbial fertilizer had a significant and positive effect on the maize plants and maize yield.

### 2.8. The Effects of Replacement of Different Proportions of Chemical Fertilizer with Bacillus sp. SYM-4 Microbial Fertilizer on the Nutrient Content of Maize Kernels

Analysis of maize kernels dried to a constant weight showed that the application of fertilizer containing certain proportions of *Bacillus* sp. SYM-4 microbial fertilizer was able to promote the growth of the maize panicles (Figure 6A,B). Plants treated with fertilizer in which 40% or 60% of the chemical fertilizer was replaced by *Bacillus* sp. SYM-4 showed significant increases in soluble protein content in the kernels, reaching 97.12 ± 2.14 mg/g and 97.21 ± 0.77 mg/g, respectively (Figure 6C). Where 20% of the chemical fertilizer was replaced by *Bacillus* sp. SYM-4 microbial fertilizer, the soluble sugar concentration in the maize kernels increased significantly (31.95 ± 0.45%) compared with the control group (Figure 6D).

The nutrients in the kernels were next determined following the application of different proportions of the commercial microbial fertilizer. Where the fertilizer applied comprised 100% *Bacillus* sp. SYM-4 microbial fertilizer, the soluble protein contents in the maize kernels were significantly higher than those obtained following application of a fertilizer comprising 100% commercial microbial fertilizer (*p* ≤ 0.001), an increase of 16.68 ± 1.54% over the control. When 20% of the chemical fertilizer was replaced by *Bacillus* sp. SYM-4 microbial fertilizer, the concentration of soluble sugars in the resulting maize kernels was significantly higher than when 20% commercial microbial fertilizer was used (*p* ≤ 0.001). The concentration was significantly increased by 22.10 ± 0.10% and 27.04 ± 7.29% (Appendix A).

## 3. Discussion

### 3.1. Bacillus sp. SYM-4 Is a PGPR for Maize

Achieving high crop yields under the premise of sustainable development has become a key issue and urgently needs to be addressed [21]. PGPR in soil, of which there are many, play a crucial role in plant growth [22,23]. Studies have demonstrated that *Bacillus* spp. have a significant effect on the promotion of plant growth [24]. For example, *Bacillus velezensis* BTR11 is able to promote the growth of rice, while *Bacillus subtilis* SL-44 can promote the growth of radish [25,26]. In this study, we isolated and purified 36 strains of bacteria from the rhizospheric soil of wild maize using a dilution plate method. These strains were then inoculated into the roots of maize seedlings. Compared with the unfertilized control group, the plant height and root length of maize seedlings inoculated with *Bacillus* sp. SYM-4 increased significantly, indicating that *Bacillus* sp. SYM-4 can also promote the growth of maize plants. *Bacillus* sp. SYM-4 was therefore identified as a PGPR of maize.

### 3.2. The Ability of Bacillus sp. SYM-4 to Synthesize IAA, Thereby Increasing the Chlorophyll Content and Enzyme Activity of Maize Seedlings, Is the Key Reason for Its Promotion of Maize Growth

PGPR can promote plant growth through a variety of mechanisms, including the production of phytohormones and increasing plant defenses against pathogens [27]. Phytohormones play a vital role in coordinating plant growth and defenses [28]. IAA is an important phytohormone and is involved in almost all aspects of plant growth and development, as well as the defense responses [29]. Indeed, the ability to produce IAA is considered to be an important indicator for the early screening of PGPR [30]. This is because the promotion of plant growth by PGPR can be achieved by bacterial synthesis of phytohormones related to growth [31]. For example, *Sphingomonas sediminicola* Dae20 has the ability to synthesize phytohormones that promote the growth of Ara-bidopsis thaliana [32]. In this study, we used UPLC-MS/MS to qualitatively and quantitatively analyze the ability of maize rhizosphere bacteria to synthesize IAA. The results showed that *Bacillus* sp. SYM-4 had the strongest ability of the isolated strains to synthesize IAA. In addition, through the qualitative and quantitative analysis of IAA contents in the aerial parts and roots of maize inoculated with *Bacillus* sp. SYM-4, we found that the IAA concentrations in the aerial parts and roots of maize increased significantly following inoculation with *Bacillus* sp. SYM-4. Currently, studies have shown that IAA can stimulate plant root elongation and lateral root formation. The enhancement of root growth is related to the increase in the availability of nutrients such as nitrogen and magnesium, which are necessary for chlorophyll synthesis. Plant crops with long green retention periods have higher yields [33]. In our study, we measured the chlorophyll content in maize leaves after inoculation with *Bacillus* sp. SYM-4 and found that the chlorophyll content in seedlings inoculated with *Bacillus* sp. SYM-4 increased significantly. In addition, IAA has been shown to activate the antioxidant defense system by regulating reactive oxygen scavenging enzymes. For example, the *TrIAA27* gene is involved in scavenging reactive oxygen species (ROS) and reducing oxidative damage, thereby promoting the growth of rapeseed [34]. Our study showed that the content of SOD, POD, and CAT in maize seedlings increased significantly after inoculation with *Bacillus* sp. SYM-4, thereby reducing oxidative damage.

### 3.3. The Beneficial Effect on Maize of Replacing a Proportion of Chemical Fertilizer with a Bacillus sp. SYM-4 Microbial Fertilizer

Maize is one of the three most important food crops globally, but the plants require a lot of fertilization. The excessive use of chemical fertilizers has caused a series of environmental problems, including soil degradation and water pollution [29]. Replacing a certain amount of chemical fertilizer with microbial fertilizer can reduce the use of chemical fertilizer and thus reduce the impact of agriculture on the environment [35]. Microbial fertilizers are nutritionally rich and can significantly improve soil quality as well as increase crop yield. However, the application of microbial fertilizer alone, or only a single application, is not effective, and any impact on plant growth is uncertain and slow to take effect [36]. A combined application of microbial fertilizer and chemical fertilizer has been found to increase soil organic matter content and accelerate the metabolism and reproduction of soil microorganisms [37]. Moreover, existing studies have shown that the survival rate and yield of watermelon plants both increased significantly following fertilization with fertilizer in which a certain proportion of the chemical fertilizer was replaced with microbial fertilizer compared with the use of chemical fertilizers alone [38]. In this study, it was found that the plant height and leaf area of maize in the treatment group following application of *Bacillus* sp. SYM-4 microbial fertilizer and chemical fertilizer were significantly higher than those in the pure chemical fertilizer group. Furthermore, it is extremely important to find a ratio of microbial and chemical fertilizers that both promotes maize growth and is environmentally sustainable. Therefore, by applying different proportions of *Bacillus* sp. SYM-4 microbial fertilizer and chemical fertilizer on maize, we found that in plants treated with a fertilizer in which 20% chemical product had been replaced with *Bacillus* sp. SYM-4 microbial fertilizer, the yield and panicle length were significantly higher and the length of the bald tip was significantly reduced compared with those in plants treated with pure chemical fertilizer. However, the maize variety used in these experiments was B73, which is not used in large-scale agriculture. Future research should investigate the use of different proportions of *Bacillus* sp. SYM-4 microbial fertilizer on different maize varieties.

## 4. Materials and Methods

### 4.1. Collection of Maize Rhizosphere Soil

Maize rhizosphere soil samples were collected from the medium-frequency transformer loam experimental field of Shenyang Agricultural University (North Temperate Continental Monsoon Climate) in July 2023. A population of maize plants has been established in this experimental field and continues to grow and reproduce every year (E: 123°57′, N: 41°82′). The fresh soil samples had an average pH range of 5.16–6.44, a total nitrogen range of 0.91–1.12 g/kg, a total phosphorus range of 0.81–1.10 g/kg, and a total potassium range of 16.45–16.98 g/kg. In order to collect samples, the whole maize plant was uprooted, the bulk soil around the root was shaken off, and the rhizospheric soil within 5 cm of the root was collected [39]. The collected soil samples were immediately brought back to the laboratory in an ice box.

### 4.2. Isolation and Purification of Bacteria from Soil

Bacteria were isolated from the soil samples using a modified dilution coating method [40]. A total of 1 g fresh rhizospheric soil was added to 9 mL sterile water and shaken at 200 rpm for 20 min at 37 °C and then allowed to stand for 20 min to precipitate the soil particles. The supernatant was continuously diluted with distilled water to 10^−1^, 10^−2^, 10^−3^, 10^−4^, 10^−5^, and 10^−6^ g/mL. Then, 100 μL of supernatant was pumped onto a lysogeny broth (LB) solid medium plate and scraped evenly over the surface with a coating device. The plate was cultured at 37 °C for 36 h, and the growth of bacterial colonies was observed. Individual colonies were then isolated and cloned strains were cultivated.

### 4.3. Strain Identification

Morphological observation and molecular identification of the isolated strains were carried out. The monoclonal morphology of the purified bacteria was observed under a microscope. Monoclonal colonies were selected and inoculated onto LB solid medium using streak culture, and the morphology was observed under an upright microscope. Bacterial DNA was extracted using a lysozyme SDS phenol/chloroform method, and PCR with the universal primers 27F and 1492R was used to amplify 16s rDNA. The PCR products were then sequenced (Sangon Biotech, Shanghai, China) and identified with BLAST searches in the NCBI database. Finally, a neighbor-joining phylogenetic tree was constructed in MEGA 7.0 [41].

### 4.4. Determination of IAA Concentration

The bacteria isolated and purified from the soil were inoculated into 100 mL liquid LB and cultured at 37 °C. Each 12 h from 12 h to 48 h after inoculation, 50 mL bacterial solution was extracted and subjected to three subsequent ethyl acetate extractions. The ethyl acetate was then evaporated from each sample using a rotary evaporator, and samples were then re-dissolved in 1 mL of chromatographic methanol for UPLC-MS/MS analysis to determine the concentration of IAA in the sample. Three biological replicates were performed in each group of experiments. Quantitative analysis was then performed using the MRM mode of UPLC-MS/MS (instrument: Shimadzu LCMS-8050, Kyoto, Japan) [18].

### 4.5. Determination of Bacterial Movement

Single colonies were picked and cultured in LB liquid medium for 12 h, then 1 mL was taken and centrifuged at 8000 rpm at 4 °C using a high-speed centrifuge. After being washed twice with sterile water, 1 mL sterile water was added, and samples were mixed well to obtain a bacterial suspension. The cells were inoculated into swimming medium and were cultured at 37 °C for 20 h. The diameter of the turbid area formed by the migration of bacteria from the inoculation point was observed and measured, and the distance moved by the bacteria was recorded. Each group of experiments was repeated three times [42].

### 4.6. Cultivation of Maize and Inoculation of Bacillus sp. SYM-4 in the Laboratory

Maize seed B73 was disinfected with sterile water for 2 min, then 75% ethanol for 5 min, and finally 10% sodium hypochlorite aqueous solution for 10 min. Samples were then washed repeatedly with sterile water until odorless, placed in a sterile culture dish containing sterile water, and germinated in a constant-temperature incubator at 25 °C.

The roots of germinated maize seedlings (about 2 cm) were immersed in a suspension of *Bacillus* sp. SYM-4 (OD_600_ about 0.3) for 1 h, with the control group being immersed in pure LB for the same length of time. Then, the maize seedlings were transplanted into a separate pot containing thrice-sterilized brown soil. Each treatment comprised 30 plants, and the experiment was repeated 3 times. The seedlings were cultivated in greenhouses (16 h of light and 8 h of dark). After 7 days, plant specimens and rhizosphere soil were collected for subsequent experiments. The length and fresh weight of buds and roots were recorded. The formula for calculating the length or weight change of shoots or roots after treatment was (T − C)/C [18], where C is the average value of the control group and T is the average value of the group inoculated with *Bacillus* sp. SYM-4.

### 4.7. Determination of IAA Concentration in Maize

A 0.5 g sample of the aerial parts or roots of maize seedlings grown for 7 d or 14 d following inoculation with *Bacillus* sp. SYM-4 was added to a 10 mL centrifuge tube. The control group consisted of maize seedlings inoculated with LB, and each sample had 3 biological replicates. After adding 5 mL chromatographic methanol to each sample, samples were subjected to ultrasonic extraction for 40 min. Following centrifugation, the remaining residue was mixed with 5 mL chromatographic methanol for a further ultrasonic extraction for 40 min. The two extraction supernatants were then combined and concentrated with rotary evaporation. Then, 5 mL methanol was added to the samples prior to evaporation to wash the wall of the bottle. After rotary evaporation, 1 mL chromatographic methanol was added. After filtration through a 0.22 μm filter membrane, the volume was adjusted to 1 mL with methanol, and the samples were qualitatively and quantitatively examined using UPLC-MS/MS in MRM mode [43]. Three biological replicates were performed for each group of experiments.

### 4.8. Determination of Leaf Area at the Four Growth Stages of Maize

The plant height and leaf area of maize seedlings at the four stages of growth were measured. The first stage is the seedling stage, which is mainly characterized by having a plant height of 20–30 cm, 3–5 leaves, and showing initial root formation. The second stage is the jointing stage. The main characteristics are obvious elongation of the basal internodes and a rapid increase in plant height. In the third stage, the large trumpet stage, the top leaves of the maize seedling form a trumpet-like structure, and the tassel begins to differentiate. Finally, 60–70 days after sowing, the maize seedlings enter the tasseling stage, and the tassel can be pulled out from the top leaf [44]. The leaf area of seedlings at each of these four stages was measured using a handheld maize leaf area measuring instrument (YMJ-A, Tuopu Yunnong, Hangzhou, China).

### 4.9. Determination of Maize Chlorophyll Content, Soluble Protein Content, and Field Maize Yield

The chlorophyll concentration in maize leaves was measured using a chlorophyll measuring instrument (SPAD-502; Konica Minolta, Tokyo, Japan). All leaves on the same plant were measured, and the average value was taken as the chlorophyll concentration. A 1 g leaf or grain sample was then added into a mortar, and PBS was added for grinding, to a volume of 10 mL. Samples were then centrifuged at 10,000 rpm for 20 min, and the supernatant was removed and analyzed. The soluble protein content in the leaves was determined using a Coomassie brilliant blue method soluble protein quantitative kit (Applygen, Nanjing, China).

The maize yield was calculated using the following formula [45]:$$\mathrm{Yield}=\frac{\mathrm{Plant}\;\mathrm{per}\;\mathrm{mu}\times \mathrm{Kernels}\;\mathrm{per}\;\mathrm{ear}\times 1000-\mathrm{Kernal}\;\mathrm{Weight}(g)}{15,000,000}$$

### 4.10. Determination of Antioxidant Enzyme Activity and Lipid Peroxidation Markers in Potted Maize Seedlings Grown Following Inoculation with Bacillus sp. SYM-4

B73 maize seedlings were grown in flowerpots for 14 d. A 0.5 g leaf sample from each seedling was then weighed and mixed with 2 mL phosphate buffer. Samples were ground into slurry and were then diluted to 10 mL with phosphate buffer. A 5 mL aliquot was centrifuged at 1000 rpm for 10 min, with the obtained supernatant being the crude extract. CAT (catalase), POD (peroxidase), SOD (superoxide), and MDA (malondialdehyde) levels were measured using CAT kits (Solarbio, Shenyang, China), POD kits (Solarbio, Shenyang, China), SOD kits (Sangon, Shanghai, China), and MDA kits (Sangon, Shanghai, China), respectively [46].

### 4.11. Field Cultivation of Maize and Preparation of Bacillus sp. SYM-4 Microbial Fertilizer

The field experiments took place at the experimental field of Shenyang Agricultural University, and variety B73 maize seeds were selected for this experiment. To prepare the field, fallow land with weeds was repeatedly plowed, and a flat and uniformly sized plot was selected. The seeds were sown at a depth of 7–10 cm.

A carrier mixture of peat, charcoal, and flower soil at a ratio of 2:2:1 was prepared and placed in a sterilization pot and subjected twice to continuous sterilization. After sterilization, the carrier was placed on an ultra-clean bench, cooled, and packaged into a fresh-keeping bag. Then, 200 mL of *Bacillus* sp. SYM-4 suspension (OD_600_ about 0.3) was added to the carrier and mixed in completely [47]. The carrier was then cultured in an incubator at 37 °C for 7 d to ensure that it remained loose and not agglomerated so as to facilitate the colonization of the strain. This colonized carrier became microbial fertilizer containing *Bacillus* sp. SYM-4.

In the following experiments, the microbial fertilizer was used to replace different proportions of chemical fertilizer (Stanley fertilizer, composed of N, P, K, and minor trace elements) at a substitution ratio of 20%, 40%, 60%, 80%, or 100%. The microbial fertilizer and chemical fertilizer were applied to opposing sides of the soil surrounding the maize plants. Fertilizer is usually applied to maize plants at the seedling stage, big trumpet stage, and filling stage. In each case, the maize was topdressed according to the plant. At the seedling and filling stages, 1.5 g of fertilizer was applied to each plant. For example, where 20% chemical fertilizer was replaced with *Bacillus* sp. SYM-4 microbial fertilizer, 1.2 g of chemical fertilizer and 0.3 g of *Bacillus* sp. SYM-4 microbial fertilizer were applied. At the big trumpet stage, maize needs a large amount of fertilizer, and a total of 3 g fertilizer was applied to each plant at this stage. Using the above example, where 20% chemical fertilizer was replaced with *Bacillus* sp. SYM-4 microbial fertilizer, 2.4 g chemical fertilizer and 0.6 g *Bacillus* sp. SYM-4 microbial fertilizer were applied (Table 2). Five biological replicates were made in both the treatment and control groups.

These experiments were repeated using a commercially available *Bacillus subtilis* microbial fertilizer (Pathfinder Pioneer). In these experiments, the maize at different growth stages, the experimental fields, and the proportions of chemical fertilizer replaced with microbial fertilizer were the same as described above. Five biological replicates were made in both the treatment and control groups.

### 4.12. Data Analysis

All data were analyzed in SPSS 22.0 and were plotted and visualized in GraphPad Prism 9.5. If the data were normally distributed, independent-samples *t*-tests were used to compare the two groups. Tukey’s test was used for analysis of variance. When *p* ≤ 0.05, the differences between groups were considered to be statistically significant [48].

## 5. Conclusions

This study identified *Bacillus* sp. SYM-4 as a novel PGPR from maize rhizosphere soil, demonstrating its superior capacity for IAA synthesis compared to other isolates. Inoculation with SYM-4 significantly elevated endogenous IAA levels and enhanced CAT and SOD activities in maize seedlings, synergistically promoting biomass accumulation. Field trials established that substituting 20% chemical fertilizer with SYM-4-based biofertilizer optimized yield parameters (kernel rows, 100-grain weight, and total yield), outperforming both full-chemical and commercial biofertilizer regimens by 32.89 ± 0.94% and 22.10 ± 0.10%, respectively. This study provides a scientific basis for the use of a certain ratio of chemical fertilizer and microbial fertilizer for the green planting of maize. Follow-up research should be extended to include more maize varieties with the aim of improving the sustainable development of agriculture and protection of maize and soil resources.

## Figures and Tables

**Figure 1 plants-14-01587-f001:**
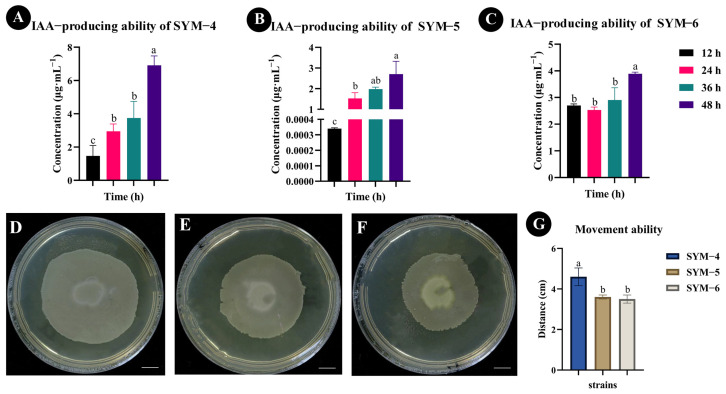
Determination of IAA-producing ability and movement ability of the strains. (**A**–**C**) Qualitative and quantitative analyses of IAA-producing ability of different bacterial strains. (**D**–**F**) Qualitative diagram of the movement ability of SYM-4, SYM-5, and SYM-6. (**G**) Quantitative analysis of the movement ability of SYM-4, SYM-5, and SYM-6. Scale bar = 1 cm. The different small letters represent significant differences at the *p* < 0.05 level.

**Figure 2 plants-14-01587-f002:**
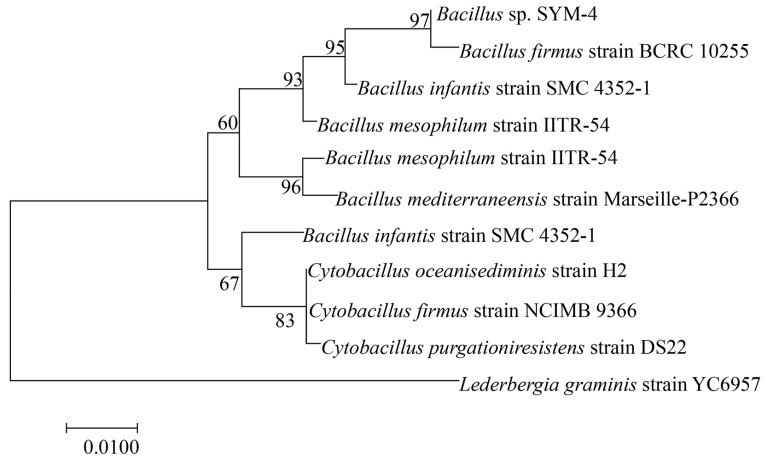
Phylogenetic tree reconstructed using DNA sequences from *Bacillus* sp. SYM-4 and sequences downloaded from NCBI.

**Figure 3 plants-14-01587-f003:**
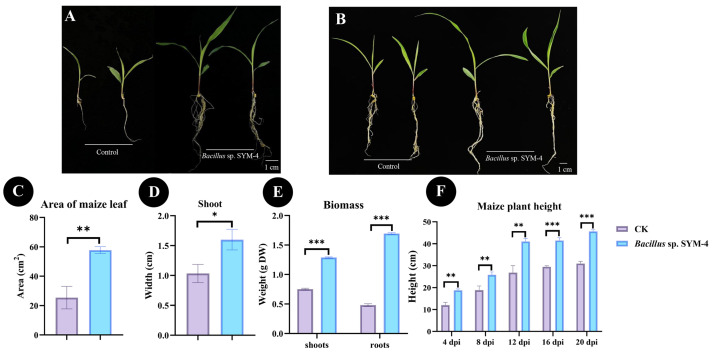
Growth promotion effects of *Bacillus* sp. SYM-4 inoculation on potted maize seedlings. (**A**) Effects of SYM-4 on the growth of maize seedlings at 7 dpi. (**B**) Effects of SYM-4 on the growth of maize seedlings at 14 dpi. (**C**–**E**) Quantitative parameters, including leaf area, stem diameter, and biomass, measured at 14 dpi. (**F**) Temporal dynamics of plant height from 4 dpi to 20 dpi. Triple asterisks indicate significant differences between the control group and other treatments, *** *p* < 0.001, ** *p* < 0.05, * *p* < 0.01.

**Figure 4 plants-14-01587-f004:**
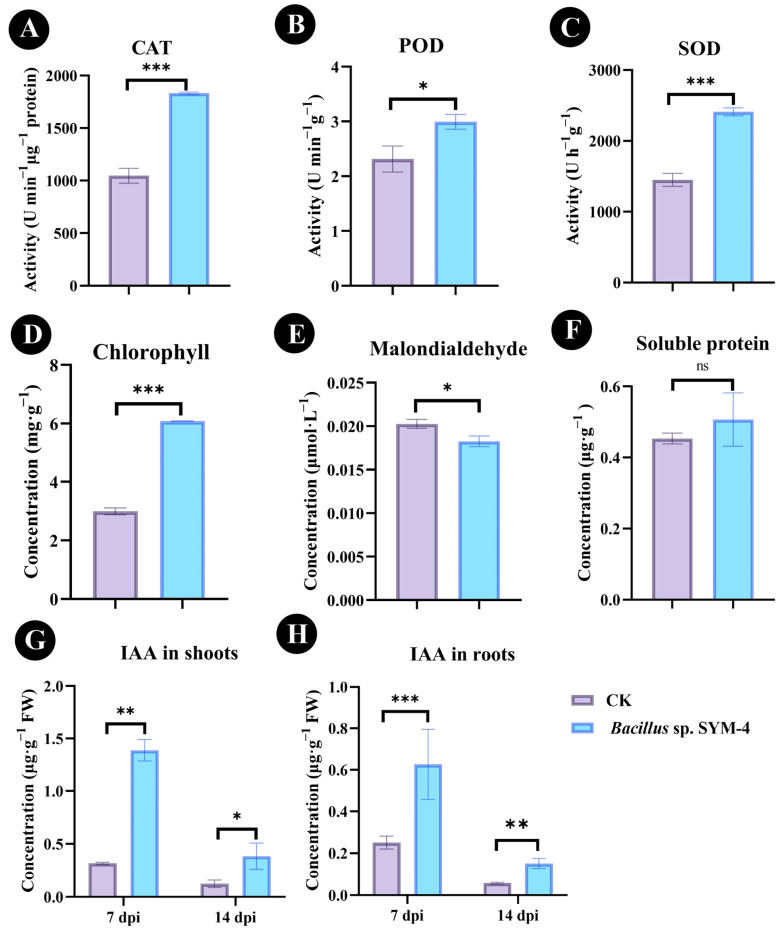
Effect of inoculation with *Bacillus* sp. SYM-4 on physiological and biochemical parameters in maize seedlings. (**A**–**C**) Effect of SYM-4 on the antioxidant enzyme activity of maize seedlings at 14 dpi. (**D**–**F**) Effect of SYM-4 on the chlorophyll content, malondialdehyde, and soluble protein of maize seedlings at 14 dpi. (**G**) Effect of SYM-4 on IAA content in shoots of maize seedlings at 7 dpi and 14 dpi. (**H**) Effect of SYM-4 on IAA content in roots of maize seedlings at 7 dpi and 14 dpi. Triple asterisks indicate significant differences between the control group and other treatments, *** *p* < 0.001, ** *p* < 0.05, * *p* < 0.01.

**Figure 5 plants-14-01587-f005:**
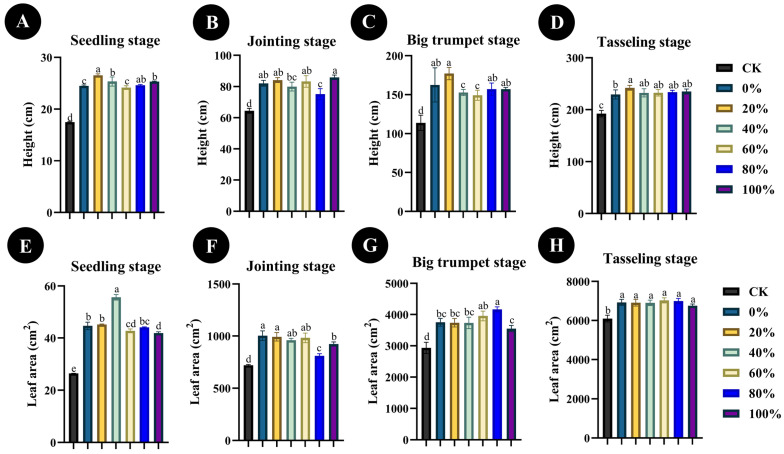
Plant height (**A**–**D**) and leaf area (**E**–**H**) of maize seedlings at four growth stages in the field, following application of fertilizer, where different proportions (0%, 20%, 40%, 60%, 80%, or 100%) of chemical fertilizer were replaced with microbial fertilizer (SYM-4). The legend shows the percentage of chemical fertilizer replaced by microbial fertilizer in each treatment. Mean differences were compared using one-way ANOVA with Tukey’s test. The different small letters represent significant differences at the *p* < 0.05 level. The results shown represent the means ± standard deviation.

**Figure 6 plants-14-01587-f006:**
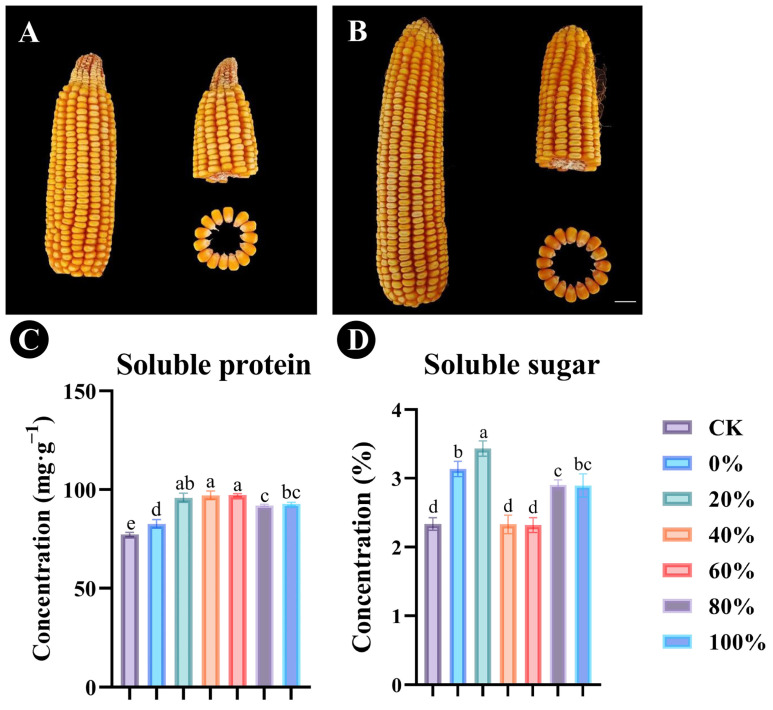
The effects of replacement of different proportions of chemical fertilizer with microbial fertilizer on maize kernels and panicles. The effects of control group (**A**) and 20% *Bacillus* sp. SYM-4 microbial fertilizer replacement of chemical fertilizer (**B**) on corn. The concentrations of soluble protein (**C**) and soluble sugars (**D**) in the maize kernels were determined. In panels C and D, the legend shows the proportions (0%, 20%, 40%, 60%, 80%, or 100%) of chemical fertilizer replaced by microbial fertilizer. Scale bar = 1 cm. Mean differences were compared using one-way ANOVA with Tukey’s test. The different small letters represent significant differences at the *p* < 0.05 level. The results shown represent the means ± standard deviation.

**Table 1 plants-14-01587-t001:** Effects of the replacement of different proportions of chemical fertilizer with *Bacillus* sp. SYM-4 microbial fertilizer on ear characters and yield of maize plants in the field. Triple asterisks indicate significant differences between the control group and other treatments, *** *p* < 0.001, ** *p* < 0.05, * *p* < 0.01.

	0%	20%	40%	60%	80%	100%	Control Group
Spike length (cm)	19.88 ± 2.43 **	20.50 ± 1.13 ***	19.40 ± 1.71 **	21.11 ± 2.23 ***	19.85 ± 2.47 **	18.52 ± 2.74 *	14.16 ± 2.44
Spike diameter (cm)	4.99 ± 0.21 ***	4.96 ± 0.15 ***	4.92 ± 0.17 ***	4.98 ± 0.11 ***	4.80 ± 0.31 **	4.75 ± 0.23 **	4.12 ± 0.24
Bare top length (cm)	1.15 ± 0.83 **	1.02 ± 0.89 ***	1.37 ± 0.54 *	1.16 ± 0.54 **	1.15 ± 0.83 **	1.16 ± 0.92 **	3.35 ± 0.37
Rows of kernels (lines)	15.11 ± 1.50 **	15.90 ± 1.88 **	16.36 ± 1.75 ***	16.13 ± 0.22 ***	15.82 ± 1.89 **	15.64 ± 1.50 **	14.67 ± 2.00
Number of kernels per row (grains)	38.06 ± 4.62 **	39.92 ± 3.48 ***	37.30 ± 3.43 **	37.28 ± 0.33 **	36.55 ± 5.96 **	34.73 ± 7.64 *	21.89 ± 5.37
100-seed weight (g)	33.06 ± 4.62 **	34.49 ± 0.38 ***	33.30 ± 3.43 **	32.70 ± 0.77 *	32.76 ± 0.33 *	33.24 ± 0.55 **	31.94 ± 0.62
Maize yield (kg/mu)	756.16 ± 13.51 **	883.18 ± 13.71 ***	768.68 ± 0.78 **	733.45 ± 2.02 **	734.55 ± 0.69 **	684.14 ± 7.01 *	603.62 ± 10.68

**Table 2 plants-14-01587-t002:** The proportion of *Bacillus* sp. SYM-4 microbial fertilizer and chemical fertilizer added at different stages of maize growth.

Growth Period of Maize	Fertilizer Type	0% (g)	20% (g)	40% (g)	60% (g)	80% (g)	100% (g)
Seeding stage	Chemical fertilizer	1.5	1.2	0.9	0.6	0.3	0
	Microbial fertilizer	0	0.3	0.6	0.9	1.2	1.5
Big trumpet period	Chemical fertilizer	3	2.4	1.8	1.2	0.6	0
	Microbial fertilizer	0	0.6	1.2	1.8	2.4	3
During grain-filling period	Chemical fertilizer	1.5	1.2	0.9	0.6	0.3	0
	Microbial fertilizer	0	0.3	0.6	0.9	1.2	1.5

## Data Availability

The original contributions presented in this study are included in the article/Appendix A. Further inquiries can be directed to the corresponding authors.

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
