# Peer review of "The IAA-Producing Rhizobacterium *Bacillus* sp. SYM-4 Promotes Maize Growth and Yield"

_plants, 2025, doi:10.3390/plants14111587_

Round 1
Reviewer 1 Report
Comments and Suggestions for Authors
This
study provides a scientific basis for the use of a certain ratio of chemical fertilizer and
microbial fertilizer for the green planting of maize. Follow-up research should be extended to include more maize varieties, with the aim to improve the sustainable development of agriculture and protection of maize and soil resources.
Author Response
This study provides a scientific basis for the use of a certain ratio of chemical fertilizer and microbial fertilizer for the green planting of maize. Follow-up research should be extended to include more maize varieties, with the aim to improve the sustainable development of agriculture and protection of maize and soil resources.
Response: I sincerely thank the review experts for their recognition of this study and valuable suggestions. We fully agree with the constructive opinion that 'follow-up studies should expand more maize varieties'. As a preliminary exploration, this study does have the limitation of variety coverage. In the follow-up work, we plan to establish a multi-variety control test system, focusing on the response differences of maize varieties with different genetic characteristics to the combined application of chemical fertilizer and bacterial fertilizer, in order to construct a more universal precision fertilization model. At the same time, the team will continue to deepen the dynamic monitoring of soil microbial communities. By quantifying the coupling relationship between soil health indicators and crop growth parameters, the team will further reveal the synergistic mechanism of the fertilization model in resource protection and productivity improvement. We believe that these expanded research will provide a more solid theoretical support for the green transformation of agriculture. Thank the experts again for the affirmation and professional guidance of this research direction.
Reviewer 2 Report
Comments and Suggestions for Authors
Please read the attached file.

Author Response
The work presented focuses on the ability of isolated Basillus to synthesize IAA and partially substitute the use of chemical fertilizers. The research is interesting and up to date. It is great that the authors traced the entire ontogeny of corn and presented harvest results, corn cob photos. It is impressive. The idea of this material itself is very interesting and important. Any methods of reducing the environmental load on soils are relevant and important. In reading the manuscript, some comments and observations arose. Please take them into account.
In the Abstract part, please divide it into two parts. This will correspond to the work done. One part should describe the results of just the bacteria treatment. The other part should describe the results of a field experiment - compensation of mineral fertilizer by bacteria.
Response: We are grateful for your suggestion regarding the Abstract section. As per your recommendation, we have revised the Abstract to clearly distinguish between the results of the bacteria treatment and those of the field experiment involving the compensation of mineral fertilizer by bacteria.
Please clarify what other qualities the strain you have studied exhibits? Perhaps it has phosphate-mobilizing activity or synthesizes nitrogen? It is still necessary to expand information about the properties of bacteria. It is very important.
Response: After the isolation and purification of Bacillus sp. SYM-4, we had determined the nitrogen fixation and phosphorus solubilization ability of Bacillus sp. SYM-4 and the results showed that Bacillus sp. SYM-4 had no activity of nitrogen fixation and phosphorus solubilization.
Description of the part the results are quite sufficient. Clarify only the underdrawing captions.
Figure 1, 3, 4, 5 - clarify the captions. Fig. 1 - D, E, F – chemotaxis?
Response: Have been revised.
Fig. 3 shows a visual demonstration of the growth of treated and control plants - specify where the control is, put a ruler scale next to it. It is very important to understand the growth parameters. Indeed you have given the proportions (Scale bar = 1 cm), but it is not enough. Leaf area and plant mass are given for 14 day old plants?
Response: Thanks for your advice. According to your suggestion, the subtitles of the manuscript are modified in detail and the scale is added to the picture again.
Please clarify what you mean by the drawing - Figure S2?
Response: Potassium release activity is one of the important indicators for bacteria to promote plant growth properties. The strain was inoculated on the modified potassium feldspar medium, and the potassium-solubilizing activity of the strain was obtained by measuring the ratio of the halo size produced by the strain to the colony size.
Part of the Discussion is inadequate. Especially it is not clear why complete
replacement of minerals by bacteria is not the most effective. Why did you not discuss the components of the antioxidant system. Why did you study the chlorophyll content? All this needs to be discussed, it's important. Because a slight activation of redox metabolism could be a pre-adaptive factor in the action of these bacteria.
Please revise the manuscript. And once again carefully consider the correlation of the results with the Discussion part. The results should not be lost.
Response: We sincerely appreciate your insightful comments and suggestions regarding our manuscript. Your emphasis on the components of the antioxidant system and the rationale behind studying chlorophyll content has prompted us to carefully reconsider and enhance our work. In response to your concerns, we have thoroughly revised the manuscript. We now include a detailed discussion on the components of the antioxidant system and clarified the importance of studying chlorophyll content in the context of our research.
We believe these revisions address your concerns and strengthen the manuscript. We are grateful for your guidance in improving our work.
Reviewer 3 Report
Comments and Suggestions for Authors
This study analyzed the growth promoting effect of Bacillus sp. SYM-4 on maize and its growth promoting effect on miaze. This study has strong application value and potential impact on the promotion and application of PGPR strains in ecological agriculture. A slight revision can be published.
Q1: The language of the entire text should be improved to meet the publication standards.
Q2: The research conclusion section is a bit cumbersome, and it is recommended to further optimize it.
Comments on the Quality of English LanguageThe language of the entire text should be improved to meet the publication standards.
Author Response
This study analyzed the growth promoting effect of Bacillus sp. SYM-4 on maize and its growth promoting effect on miaze. This study has strong application value and potential impact on the promotion and application of PGPR strains in ecological agriculture. A slight revision can be published.
Q1: The language of the entire text should be improved to meet the publication standards.
Response: We sincerely appreciate the reviewer's professional guidance on language refinement. By hiring a certified editing service to standardize academic terminology and eliminate grammatical errors, the manuscript has undergone comprehensive linguistic revisions. The English language of this manuscript was revised by Dr. Jane Marczewski (A.J.M. Scientific Editing ajm-scientific-editing.com).
Q2: The research conclusion section is a bit cumbersome, and it is recommended to further optimize it.
Response: We sincerely appreciate your valuable feedback regarding the research conclusion section of our manuscript. Based on your suggestion that the section was somewhat cumbersome, we have carefully reviewed and optimized it to ensure clarity, conciseness, and logical flow. We have streamlined the content, removed redundant statements. We believe these revisions address the concerns you raised and enhance the overall quality of the manuscript. Thank you again for your guidance in improving our work.
Reviewer 4 Report
Comments and Suggestions for Authors
Please find the attachment.

Author Response
The current study aimed to isolate PGPR from maize rhizosphere soil, elucidate their growth-promoting mechanisms, and determine the optimal chemical-to-microbial fertilizer ratio through systematic evaluation. The introduction presented a clear significance and research idea about this paper. The experiment and measurements were well conducted, and the results were informative and interesting. My review of the paper is positive, after correction, the paper will be acceptable for publication.
Abstract:
Line 9-10: “microbial fertilizer” should be changed to “microbial fertilizers”; “means” should be changed to “approach”.
Response: Have been revised.
Line 21-25: “The replacement…chemical fertilizer.”, rephrasing is suggested, also pay attention to grammar.
Response: Thank you for your suggestion, we have corrected it in the abstract.
Introduction: In this part, the authors mentioned a lot of points, but they fail to adequately support the key research content of the article. For instance, the growth and development processes of maize under microbial fertilizer, etc.
Response: In the introduction, we have added sentences and references regarding the impact of microbial fertilizer on maize growth.
Line 41-42: “But the rhizosphere…many pathogens”, rephrasing is suggested.
Response: This sentence has been deleted.
Results:
Part 2.1:
- The labels for D-F in Figure 1 should be clearly marked, the authors did not mention them in this part.
Response: Have been revised.
- Figure 1G should have significant markers.
Response: Thanks for your advice. The underdrawing captions has been revised in detail according to your suggestion.
Part 2.3:
Line 116-119: “The leaf area…group”, rephrasing is suggested.
Response: Have been revised.
Line 119-121: “Indeed, seedlings…group”, rephrasing is suggested.
Response: Have been revised.
In general, the result description is very confusing and needs to be strengthened.
Response: We are truly grateful for your feedback on our manuscript. We sincerely apologize for the confusion in the results description. In response to your comments, we have revised and strengthened the results section. We hope these revisions address the concerns you raised and improve the overall quality of the manuscript.
Discussion:
“DISCUSSION” IS NOT “INTRODUCTION”. Please provide an in-depth discussion
of the experimental results. In addition, some content currently in the “Discussion” part would be more appropriately placed in the “Introduction” part.
Response: Thank you for your insightful comments on our manuscript. We sincerely apologize for the confusion between the “DISCUSSION” and “INTRODUCTION” sections. In response to your feedback, we have thoroughly revised the “DISCUSSION” section to provide a more in-depth analysis of the experimental results. We have reorganized the content to ensure that the discussion is focused, logical, and well-supported by the data.
We appreciate your guidance in helping us enhance the clarity and quality of our work.
Figures: Please provide clear notes for the content in each figure.
Response: Thanks for your suggestion regarding the clarity of the figure notes. We have now provided clear and detailed notes for each figure in the manuscript. These notes have been revised to ensure they accurately describe the content and context of each figure, enhancing the reader's understanding of the presented data.
Round 2
Reviewer 2 Report
Comments and Suggestions for Authors
The authors did a great job and took into account all my comments and remarks. All my concerns disappeared.
Author Response
Great thanks for the reviewer's contribution to improving the quality of this manuscript. Great thanks for the reviewer's recognition.
Reviewer 4 Report
Comments and Suggestions for Authors
Well done, this version is better than before. I I only have one suggestion, “DISCUSSION” IS NOT “INTRODUCTION”. The authors list a lot of previous research findings in the discussion part, but provide little analysis of their own results, such as part 3.3.
Author Response
Well done, this version is better than before. I only have one suggestion, “DISCUSSION” IS NOT “INTRODUCTION”. The authors list a lot of previous research findings in the discussion part, but provide little analysis of their own results, such as part 3.3.
Response: Thank you very much for your positive feedback and valuable suggestion. We truly appreciate the time and effort you have devoted to reviewing our manuscript. In response to your point that the 'DISCUSSION' is not 'INTRODUCTION', and that the discussion section heavily focuses on previous research findings while analyzing our own results less, we have made modifications in the text. To make the revisions more clear and visible, we have changed the font color of the revised parts in the manuscript to blue. We hope this will make it easier for you to review the changes we have made. Once again, thank you for your suggestion. We believe that the revisions we have made have improved the quality and clarity of our manuscript, and we hope that you will find the revised version satisfactory.